# Impact of a Surgical Approach on Endometrial Cancer Survival According to ESMO/ESGO Risk Classification: A Retrospective Multicenter Study in the Northern Italian Region

**DOI:** 10.3390/cancers17132261

**Published:** 2025-07-07

**Authors:** Vincenzo Dario Mandato, Anna Myriam Perrone, Debora Pirillo, Gino Ciarlini, Gianluca Annunziata, Alessandro Arena, Carlo Alboni, Ilaria Di Monte, Vito Andrea Capozzi, Andrea Amadori, Ruby Martinello, Federica Rosati, Marco Stefanetti, Andrea Palicelli, Giacomo Santandrea, Renato Seracchioli, Roberto Berretta, Lorenzo Aguzzoli, Federica Torricelli, Pierandrea De Iaco

**Affiliations:** 1Unit of Obstetrics and Gynecologic Oncology, Azienda USL-IRCCS di Reggio Emilia, 42123 Reggio Emilia, Italy; debora.pirillo@ausl.re.it (D.P.); gino.ciarlini@ausl.re.it (G.C.); gianluca.annunziata@ausl.re.it (G.A.); lorenzo.aguzzoli@ausl.re.it (L.A.); 2Division of Oncologic Gynecology, IRCCS Azienda Ospedaliero-Universitaria di Bologna, 40138 Bologna, Italy; cmyriam.perrone@aosp.bo.it (A.M.P.); pierandrea.deiaco@unibo.it (P.D.I.); 3Department of Medical and Surgical Sciences, University of Bologna, 40138 Bologna, Italy; calessandro.arena@unibo.it (A.A.); renato.seracchioli@unibo.it (R.S.); 4Division of Gynaecology and Human Reproduction Physiopathology, IRCCS Azienda Ospedaliero—Universitaria di Bologna, 40138 Bologna, Italy; 5Department of Medical and Surgical Sciences for Mother, Child and Adult, University of Modena and Reggio Emilia, 41124 Modena, Italy; carlo.alboni@aou.mo.it (C.A.); ilaria.dimonte@aou.mo.it (I.D.M.); 6Department of Obstetrics and Gynecology, University of Parma, 43121 Parma, Italy; vitoandrea.capozzi@ao.pr.it (V.A.C.); roberto.berretta@unipr.it (R.B.); 7Gynecology Unit, Ospedale di Forlì, 47121 Forlì, Italy; dottamadori@gmail.com; 8Section of Obstetrics and Gynecology, Department of Medical Sciences, University of Ferrara, 44124 Ferrara, Italy; ruby.martinello@unife.it; 9Gynecological and Obstetrical Unit, Infermi Hospital, 47923 Rimini, Italy; federica.rosati@auslromagna.it (F.R.); marco.stefanetti@auslromagna.it (M.S.); 10Pathology Unit, Azienda USL-IRCCS di Reggio Emilia, 42123 Reggio Emilia, Italy; andrea.palicelli@ausl.re.it (A.P.); giacomo.santandrea@ausl.re.it (G.S.); 11Laboratory of Translational Research, Azienda USL-IRCCS di Reggio Emilia, 42123 Reggio Emilia, Italy; federica.torricelli@ausl.re.it

**Keywords:** adjuvant therapy, endometrial cancers, laparoscopy, laparotomy, survival, lymphadenectomy, sentinel lymph node biopsy, ESMO-ESGO classification system, high-risk endometrial cancer

## Abstract

Endometrial cancer (EC) is the most common gynecological tumor in Western countries; in recent decades, the laparoscopic approach has been proposed as the gold standard. After the publication of the study “Laparoscopic Approach to Carcinoma of the Cervix”, which showed a worsening of oncological outcomes in patients treated with laparoscopy, doubts have emerged about the safety of laparoscopy also in EC treatment. The real-world data of this study demonstrate how the use of laparoscopy has increased over the years and has proven to be a safe approach in EC treatment, regardless of the risk class. Indeed, laparoscopy was associated with a better prognosis in patients undergoing CE, especially in those at high risk of recurrence.

## 1. Introduction

Endometrial cancer (EC) is the most common malignancy of the female genital tract in Western and emerging countries. In 2022, 420,242 new cancer cases and 97,704 cancer deaths were diagnosed worldwide [1]. Most EC patients are identified in the early stages, largely because symptoms tend to appear soon after disease onset. As a result, the prognosis is typically positive, with a 5-year overall survival rate of 77% [2]. In contrast, outcomes are significantly worse for patients with advanced or recurrent EC, mainly due to the limited effectiveness of chemotherapy [3,4,5,6].

Endometrioid tumors represent the most common EC and they are caused by unopposed hyperestrogenism, are typically diagnosed at an early stage, and have a good prognosis. Non-endometrioid ECs are rarer and more aggressive. Patient’s age, International Federation of Gynaecology and Obstetrics (FIGO) stage, depth of myometrial invasion, tumor histotype, grade (G), and lymphovascular space invasion (LVSI) are well known prognostic factors [7,8]. Over the past decade, new risk factors have been studied to better predict the risk of recurrence [9,10,11]. Recent molecular classifications have stratified EC into the following four distinct genomic subgroups: ultra-mutated DNA polymerase ɛ (POLE), which is associated with an excellent prognosis; microsatellite instability hypermutated (MSI-H) and copy-number low (microsatellite stable) tumors, both linked to intermediate clinical outcomes; and copy-number high (serous-like) tumors, characterized by an unfavorable prognosis. Notably, a subset of high-grade endometrioid carcinomas show genomic and mutational profiles very similar to those of serous ECs. Excluding POLE mutations, no single genetic alteration has been identified that uniquely defines any of the four molecular subtypes [12,13,14,15,16,17].

However, in some countries most hospitals do not perform molecular analysis due to lack of resources and lack of surrogate markers for POLE immunochemistry [13,18,19,20,21]. The main EC treatment is total hysterectomy with bilateral salpingo-oophorectomy with/without lymph node assessment [7,22]. Laparoscopy (LPS) is the recommended approach particularly in the low-risk patients [22,23,24]. Two randomized controlled trials demonstrated that LPS was safe and associated with better perioperative outcomes and quality of life compared with laparotomy (LPT) [22,23]. Despite only 17% of patients included being at high-risk, LPS was suggested also in G3 EC since detrimental effects were not reported [12,22,23]. After the publication of the results of the LACC trial, which shocked the scientific community [25,26,27,28], gynecologists once again questioned the safety of LPS also in the treatment of EC. Cervical cancer patients treated with minimally invasive surgery presented a higher recurrence rate and worse overall survival than patients treated with LPT [25], and so did EC patients treated with LPS also have the same risk? Considering the short follow-up period of the patients enrolled in the two trials, some doubts arise about the long-term oncological outcomes in patients treated with LPS [29,30]. In this study we compared the different impacts of the surgical approach on the survival of EC patients stratified according to the 2016 ESMO-ESGO recurrence risk classification system [24].

## 2. Methods

In accordance with the journal’s guidelines, we will provide our data for independent analysis by a team selected by the Editorial Team for the purposes of additional data analysis or for the reproducibility of this study in other centers if such is requested.

### 2.1. Study Design

All patients who underwent a hysterectomy for EC in seven hospitals of the northern Italian region Emilia Romagna, from 2000 to 2019, were included in the study (Azienda USL-IRCCS di Reggio Emilia, Reggio Emilia, Italy. University of Bologna, Bologna, Italy. University of Modena and Reggio Emilia, Modena, Italy. University of Parma, Parma, Italy. University of Ferrara, Ferrara, Italy. Ospedale di Forlì, Forlì, Italy. Ospedale degli Infermi, Rimini, Italy). Patients younger than 18 years and with concurrent malignancy were excluded. All cases were revised and staged according to the 2009 International FIGO staging system [31]. Subsequently, each patient was stratified based on the surgical approach and the 2016 ESMO-ESGO recurrence risk classification system [24].

### 2.2. Data Collection

Patients were identified by record linkage between data retrieved from hospital records, the pathological database, and oncological and gynecological follow-up visits. Clinical and pathological data were recorded for each patient included. Patients’ characteristics, including age, parity, body mass index (BMI), American Society of Anaesthesiologists (ASA) classification system score, menopausal status, symptoms and comorbidities such as diabetes and hypertension, and use of hormone replacement therapy (HRT), tamoxifen (TMX), and other therapies, were recorded. Use of diagnostic hysteroscopy (DH), dilatation and curettage (D&C), computed tomography scan (CT), transvaginal ultrasound (TVUS), magnetic resonance (MRI), X-ray (XR), positron-emission tomography (PET), vaginal hysterectomy, LPS, LPT, peritoneal biopsy, peritoneal washing, pelvic lymph node dissection (PLND), paraaortic lymph node dissection (PALND), sentinel lymph node dissection (SLND), total lymph node retrieved, number of positive lymph node, duration of surgery, hospital length of stay (LoS), FIGO stage, hystology, grade, LVSI, class of risk, adjuvant treatment, recurrence, site of recurrence, death, total survival, disease free survival (DFS), and overall survival (OS) were reported. Complications such as postoperative fever, hemoglobin variation, and a requirement for blood transfusions were also reported.

The analysis was conducted using all available data without imputation. For each variable, the number of missing observations was documented (Table 1). The total cohort presented a median follow up period of 46.5 months (IQR = 60 months) and the sub-cohort treated with laparoscopy presented a median follow up of 33 months (IQR = 40 months), whilst the sub-cohort treated with laparotomy presented a median follow up of 64 months (IQR = 88.5 months).

### 2.3. Statistical Analysis

All statistical analyses were conducted using R software version 4.3.1 (R Foundation for Statistical Computing, Vienna, Austria). Analysis of association was performed by applying Fisher’s exact test for categorical variables and the Kruskal–Wallis test for the comparison of continuous variables between two groups. OS was calculated as the period spent from the treatment date to the date of death or last follow-up. Survival analyses were represented by Kaplan–Meier curves using R “Survminer” package and statistical differences were evaluated by log-rank test. Multivariate survival analyses were performed by applying the Cox model. Significant statements refer to *p*-values lower than 0.05.

## 3. Results

### 3.1. Clinical Characteristics

In this study we included 2402 EC patients treated from 2000 to 2019 in seven clinical centers in Emilia Romagna. Clinical characteristics of the total cohort are summarized in Table 1.

### 3.2. Surgery

Data showed a continuous increase in the use of LPS across the years, with the use of LPS increasing from 5% in 2000 to 81% in 2019 (Figure 1).

In total, LPS was preferred in 1283 (53.6%) of patients. Excluding the small percentage of patients treated with vaginal hysterectomy (72 patients, 3%), the analyses were focused on a detailed comparison between patients treated with LPS (n = 1283) and LPT (n = 1037). LPS patients presented a significantly lower median age at diagnosis in comparison with LPT patients (65 vs 67 years, *p* < 0.001), while no significant difference was observed in BMI. Patients presenting comorbidities such as hypertension or diabetes received LPS in a lower percentage of cases in comparison with patients without comorbidities, in particular LPS was performed in 52.1% (596/1144) of patients with hypertension versus 60.3% (632/1047) of non-hypertensive patients (*p* < 0.001) and in 49.6% (177/357) of diabetic patients versus 57.3% (1045/1823) of non-diabetic ones (*p* = 0.007). Moreover, a significant reduction in the use of LPS was observed in patients with a higher ASA score, with 71.7% of ASA score I (109/152), 63.6% of ASA score II (760/1195), 50.1% of ASA score III (401/801), and only 19% of ASA score IV patients (4/21) undergoing LPS (*p* < 0.001) (Figure 2A).

More than 50% of patients at low-, intermediate-, and intermediate–high-risk EC received LPS, whilst less than 40% of patients with high-risk and advanced/metastatic tumors received LPS (*p* < 0.001). Similar data were observed concerning FIGO stage, as 61% of stage I patients (1122/1839) received LPS, whilst about 30% of patients with a higher stage were similarly treated (34% stage II, 26.9% stage III, and 38.6% stage IV) (*p* < 0.001). Also, the histology of the tumor drove the surgical choice; LPS was applied in 58.4% (1144/1958) of endometrioid EC and only in 38.5% (138/358) of non-endometrioid EC (*p* < 0.001), and between endometrioid tumors 62.1% of G1-G2 (1013/1631) underwent LPS versus 39.8% of G3 (128/322) (*p* = 0.001). No significant differences were observed concerning the presence of LVSI (Figure 2B).

LPS reduced complications and LoS. Median duration of surgery was significantly lower in patients who received LPS in comparison with LPT (140 min vs. 173 min, *p* < 0.001) (Figure 3A) and a lower percentage of LPS patients presented a fever higher than 38 °C for more than 24 h (2.5% LPS vs. 4.9% LPT, *p* = 0.004) (Figure 3B). LPS was associated with a lower reduction in hemoglobin than LPT (median: −1.5 g/dl vs. −19 g/dl, *p* < 0.001) (Figure 3C) at 24 h from operation and with a lower percentage of transfusions (2.4% LPS vs. 11.8% LPT, *p* < 0.001) (Figure 3D). In general, LoS was shorter for LPS patients compared to LPT (median: 4 vs. 7 days, respectively, *p* < 0.001) (Figure 3E).

Staging procedures were preferentially performed by LPT, in fact PLND was performed in only 47.5% (608/1283) of LPSs and in 59.9% (621/1037) of LPTs (*p* < 0.001). Similarly, PALND was performed in 13.6% (174/1283) of LPSs and 18.5% (192/1037) of LPTs (*p* = 0.001) (Figure 4A). The median number of lymph nodes retrieved was significantly different in the two surgical approaches (12 by LPS and 15 by LPT, *p* < 0.001) and collimated with the average percentage of positive lymph nodes identified (1.7% by LPS versus 5.3% by LPT, *p* < 0.001). SLND was performed in 12.3% (90/1283) of LPSs versus only 2.2% (12/1037) of LPTs (*p* < 0.001) (Figure 4A). Nevertheless, in our cohort about 50% of PLND and PALND were treated by LPS and the remaining 50% by LPT (Figure 4B,C). Conversely, LPS was the elective choice for sentinel lymph node biopsy, used in 88.2% of cases (Figure 4D).

### 3.3. Adjuvant Treatment

We evaluated the association between the surgical treatment and the following choice to perform any kind of adjuvant treatment. To obtain more informative and unbiased information we subdivided patients based on ESMO-ESGO risk class [19]. Data showed that in low-risk patients adjuvant treatment was performed after 14.1% (39/277) of LPTs and after only 9.2% (54/592) of LPSs (*p* = 0.034). Meanwhile, in intermediate–high-risk patients adjuvant treatment was delivered after LPS in 70.7% (135/192) of cases while it followed LPT in 56.4% (53/95) of patients (*p* = 0.023). Interestingly, the choice to perform adjuvant therapy was significantly more frequent after LPT than LPS (50.4% vs. 34.4%, *p* < 0.001) for endometrioid tumors, while no differences were observed in other histotypes which received adjuvant therapy in about 60% of cases independently of the surgical approach (Appendix A).

In all risk groups except for advanced/metastatic cases, patients who received adjuvant therapy previously received more complex surgeries, in which the operative time was longer, lymphadenectomy was performed in a higher percentage of cases, and a higher number of lymph nodes were analyzed. In particular, in low-risk EC the median surgical duration was 180 min for patients subsequently subjected to adjuvant therapy and 130 min for patients who did not receive the treatment (*p* < 0.001) (Appendix A). In this case no significant differences were observed in the percentage of women who underwent lymphadenectomy (Appendix A), but between patients who received staging procedures a higher number of lymph nodes were examined in those who subsequently received adjuvant therapy (median 3 vs. 10, *p* = 0.005) (Appendix A). Similarly, in the intermediate-risk group the duration of surgery was slightly higher in adjuvant-treated patients (median 164 min vs. 148 min, *p* = 0.059) and no significant differences were observed in staging procedures, but a higher number of lymph nodes were examined in patients who underwent adjuvant therapy (median 13 vs. 6, *p* = 0.025). In intermediate–high- and high-risk patients the comparison between patients treated with or without adjuvant therapy confirmed the longer median duration of surgery (intermediate–high: 165 min vs. 150 min, *p* = 0.013, high: 180 min vs. 150 min, *p* < 0.001) (Appendix A), the higher percentage of PLNDs (intermediate–high: 79.3% vs. 60.8%, *p* = 0.001, high: 76.4% vs. 57.9%, *p* < 0.001) (Appendix A), and the higher median number of lymph nodes examined (intermediate–high: 17 vs. 15, *p* = 0.081, high: 21 vs. 14, *p* < 0.001). In the high-risk group the percentage of PALNDs were also significantly higher in patients who underwent adjuvant treatment in comparison with patient who did not undergo this therapy (36.4% vs. 18.9%, *p* < 0.001) (Appendix A). These patients considered at high-risk and treated with adjuvant therapy were in a lower percentage affected by hypertension (49.5% vs. 59.6%, *p* = 0.032) or were in treatment for comorbidities (45.3% vs. 57.3%, *p* = 0.024). They presented a higher percentage of stage III tumors (49% vs. 26%) and a lower percentage of stage I tumors (36% vs. 59%) (*p* < 0.001). A total of 87.4% (313/358) of endometrioid tumors classified as high-risk were treated with adjuvant therapy versus only 64.4% (213/331) of other histotypes (*p* < 0.001). A total of 37 of 44 patients (84%) with advanced or metastatic disease received adjuvant therapy.

### 3.4. Survival

Analysis of the effect of surgical choice on OS of the total cohort evidenced a significant association between the use of LPS and a better prognosis in EC patients (Table 2, Figure 5A). A more detailed survival analysis performed by stratifying patients by ESMO-ESGO risk class showed that non-significant differences were observed in low-, intermediate-, and intermediate–high-risk patients while a significant detrimental effect of LPT on prognosis was registered in high-risk (*p* = 0.018) and advanced/metastatic patients (*p* = 0.029) (Table 2, Figure 5B,C).

Interestingly, focusing on the high-risk group, this different effect of surgical treatment on OS was maintained when also considering the combined treatment with adjuvant therapy (*p* = 0.0078) (Figure 6A). In particular, in high-risk patients treated with LPS the improvement of OS seems to be independent from the adjuvant therapy (*p* = 0.53) (Figure 6B), whilst in high-risk patients who received LPT the use of adjuvant therapy induces a significant improvement of prognosis (*p* = 0.023) (Figure 6C). On the other hand, survival analysis performed on high-risk patients treated with adjuvant therapy did not show any difference between LPS and LPT treated patients (*p* = 0.2) (Figure 6D), while the same analysis conducted on high-risk patients who did not receive adjuvant therapy showed that in this subgroup of patients LPS can significantly improve OS in comparison with the use of LPT (*p* = 0.0091) (Figure 6E).

Interestingly, the distribution of FIGO stages was significantly different (*p* < 0.001) between high-risk patients treated with LPS or LPT; between patients treated with LPS 51% were stage I, 15% were stage II, and 34% were stage III, while in the LPT group 37% were stage I, 15% stage II, and 48% stage III (Figure 7A). Moreover, focusing on non-endometrioid EC, in the LPS treated group 56% of ECs were stage IA versus 28% in LPT treated group (*p* < 0.001) (Figure 7B).

LPS was associated with improved OS in advanced/metastatic patients treated with adjuvant therapy (*p* = 0.03) (Figure 8). Only 7 of 44 advanced/metastatic patients did not receive adjuvant therapy.

To better understand the effect of therapeutic choices on overall survival (OS) and in accordance with previous analyses, multivariate Cox regression models were applied to the total cohort. The following two models were constructed: Model 1, which included risk class, surgical strategy, and adjuvant therapy; and Model 2, which added the variables age, diabetes, hypertension, and ASA score to evaluate their potential confounding effect on the associations between clinical variables and survival outcomes. As expected, age emerged as a significant negative prognostic factor for endometrial cancer. Notably, the intermediate- and intermediate–high-risk classes, which were significantly associated with worse survival in Model 1 (both HR 1.93; *p* = 0.0167 and 0.0297, respectively), lost statistical significance in Model 2. In contrast, the high-risk and advanced/metastatic groups remained strongly and significantly associated with poorer survival in both models, confirming a robust and age-independent effect (high-risk: HR = 3.77, *p* < 0.001; advanced/metastatic: HR = 18.34, *p* < 0.001). Regarding surgical approach, LPS was significantly associated with improved survival in Model 1 (HR 0.74, *p* = 0.0435), but this association was weakened and became non-significant in Model 2 (HR 0.76, *p* = 0.0750). Adjuvant therapy did not show a significant association with OS in the multivariate models applied to the total cohort. Similar dual models were constructed to assess OS in two sub-cohorts: high-risk patients and advanced/metastatic patients. For the high-risk group, the models also included FIGO stage, histological subtype, and LVSI. For the advanced/metastatic group (entirely FIGO stage IV), histotype and LVSI were added. In the high-risk sub-cohort, age remained a significant negative prognostic factor. As expected, FIGO stage III was significantly associated with worse OS compared to stage I (Model 1 HR = 1.76, *p* = 0.0364; Model 2 HR = 1.76, *p* = 0.0377), whereas stage II was not significantly different. Similarly, non-endometrioid histology was significantly associated with worse survival in both models (Model 1 HR = 1.88, *p* = 0.0039; Model 2 HR = 1.92, *p* = 0.00372), suggesting a strong and consistent prognostic effect, independent of age. LVSI was significant in Model 1 (HR = 1.58, *p* = 0.0460) but became borderline in Model 2 after adjusting for clinical variables (HR = 1.55, *p* = 0.0531), again pointing to a potential confounding role of age. The surgical approach showed a consistent and robust protective effect related to LPS compared to LPT in both models, with an approximately 50% reduction in hazard (Model 1 HR = 0.50, *p* = 0.0058; Model 2 HR = 0.48, *p* = 0.0042) indicating that the survival benefit of LPS in this subgroup is independent of age. As with the full cohort, adjuvant therapy was not significantly associated with OS in the high-risk subgroup when adjusted in multivariate models. Finally, in the advanced/metastatic sub-cohort, age was not a significant predictor of survival. The only variable consistently associated with a strong and significant improvement in OS was adjuvant therapy in both models (Model 1 HR = 0.06, *p* = 0.0013; Model 2 HR = 0.00, *p* < 0.001), indicating the robust protective effect of this treatment in patients with advanced disease (Appendix A).

## 4. Discussion

The use of LPS has increased over the years, from less than 15% to more than 80% of EC patients (Figure 1).

LPS was preferred to treat the early stages of endometrioid G1-2 EC and was preferred for the treatment of low-, intermediate-, and intermediate/high-risk patients. Although lymphadenectomy and adjuvant therapy were performed mainly in LPT patients, LPS showed no adverse effects on OS in any recurrence risk class. Particularly, in high-risk EC patients, LPS was associated with an increased OS in comparison with women treated by LPT regardless of the use of adjuvant therapy. In the literature, the use of LPS ranges from 33.6% to over 80% for EC in high-volume hospitals [32,33,34,35,36,37]. In our study LPS was used in the majority of EC patients, but 43.4% of patients underwent LPT (Table 1).

Commonly associated factors with not undergoing LPS are uterine size >12 cm and advanced stages (III-IV) [36,38]. The use of LPS varies according to histotype (use in non-endometrioid EC was 29.1–67.6%) [24,28,29,30,31], grade (use in G3 EC was 7.8-75%) [29,39,40,41], and stage (use in stage II and stage III was 9.6–51.3% and 17.9–39.4%, respectively) [32,36,37]. Usually, LPS is used in low-risk EC and LPT in high-risk patients, while in patients with intermediate- and intermediate–high-risk EC their use is similar [37]. It is well known that adequate surgical staging allows for the identification of patients requiring adjuvant therapy [32]. In the study by Vardar et al., adjuvant therapy was administered to twice as many patients treated with LPT as those treated with LPS [37]. In an Italian study, rates of stage I EC patients receiving adjuvant therapy were significantly higher in the LPS group than in the LPT group [42]. In a study by Hu et al., adjuvant CHT was administered to a significantly greater proportion of patients with clear-cell EC treated by LPT compared to those treated with LPS. However, no different oncological outcomes were found [43]. Conversely, in other studies the surgical approach did not influence the administration of adjuvant therapy [37,44]. Usually, lymphadenectomy is used to evaluate the quality of surgery. Lymphadenectomy is usually considered adequate when more than 10 nodes are removed [45,46]. Retrospective reviews showed that survival improves when at least 10–12 lymph nodes are removed during LND [47,48,49]. According to the Mayo criteria, >22 lymph nodes should be removed for adequate PLND [50]. In recent years, however, we have witnessed a paradigm shift where we have gone from the need to remove an ever-increasing number of lymph nodes to the removal of only the SNL. The SNLD technique is progressively replacing systematic lymphadenectomy. In our population, this technique was used in only the 7.7% of EC patients, and as expected, SLND was mainly performed by LPS (Figure 4D).

The final analysis of the results of the LACC trial confirmed that worse survival was associated with the minimally invasive approach; furthermore, patients treated with the minimally invasive approach more often presented peritoneal carcinomatosis at recurrence [51]. According to previous retrospective [36,37,38,52,53,54] and randomized prospective studies such as LAP2, our finding showed that LPS did not affect OS [23].

In multivariate analysis, LPS was associated with a better survival, but when clinical variables (age, diabetes, hypertension, and ASA score) were included only older age and higher recurrence risk classes influenced survival of the entire patient cohort. This finding may reflect age-related selection bias as younger patients are more likely to undergo LPS. According to previous studies, age is an independent prognostic factor. Furthermore, advanced age can often be associated with numerous comorbidities, however, comorbidities did not influence the outcome in our study [55,56].

On the contrary, considering only high-risk EC patients, LPS was associated with an increased OS regardless of age. Particularly in stage I non-endometrioid high-risk EC patients, LPS was associated with an increased OS regardless of the use of adjuvant therapy. It is known that well staged, stage I non-endometrioid EC may not receive adjuvant therapy if tumor is absent or limited to the mucosa in final histology [57]. Furthermore, in advanced/metastatic patients, adjuvant therapy alone was associated with improved OS. However, considering that, as expected, LPS reduced complications and LoS (Figure 3A–E), LPS could reduce the time to access adjuvant therapy.

LPS should always be preferred, even in cases of risk of conversion to LPT. The risk of conversion has been reported to be higher in advanced stages without affecting the prognosis, but conversion should not be considered a complication [44,58,59]. However, conversion and longer operation time has been reported to impact physical function and recovery after surgery [60]. This study has several limitations due to the retrospective nature and the long observation period, and the presence of missing data could have affected the quality of the study. Furthermore the lack of molecular data could make it obsolete. On the other hand, the multicenter nature and the large sample size have shown us what the evolution of EC treatment has been in recent decades based on real-life data. LPS can be used to treat EC of any histotype and any risk group since survival results are comparable to those of LPT. In a future study, molecular data should be reviewed to confirm this finding. Randomized controlled trials focused on patients with high-risk EC would be necessary. Considering the well-known benefits of LPS, it should always be preferred when it can guarantee oncological safety. The main goal is to perform the most oncologically correct and rigorous procedure according to current guidelines regardless of the surgeon’s preference for the approach.

## Figures and Tables

**Figure 1 cancers-17-02261-f001:**
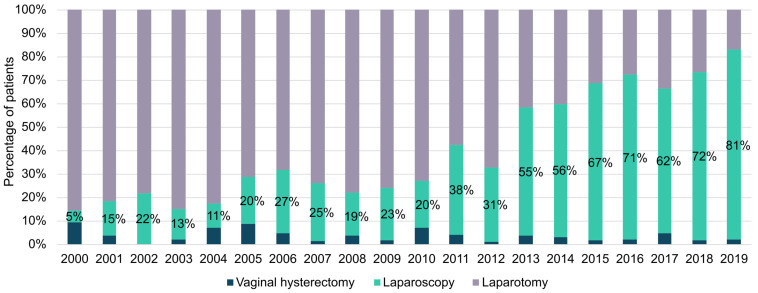
Histograms showing the frequency distribution of surgical approaches over the years.

**Figure 2 cancers-17-02261-f002:**
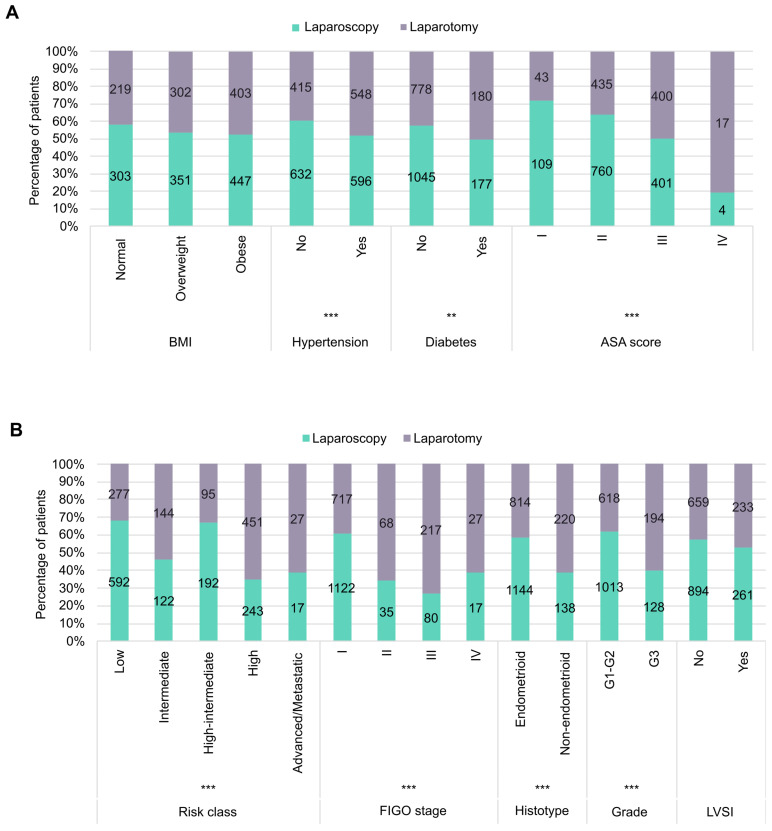
(**A**) Histograms of the percentage of patients with different clinical characteristics treated with laparoscopy or laparotomy. (**B**) Histograms of the percentage of patients with different pathological characteristics treated with laparoscopy or laparotomy. ** *p* < 0.01, *** *p* ≤ 0.001.

**Figure 3 cancers-17-02261-f003:**
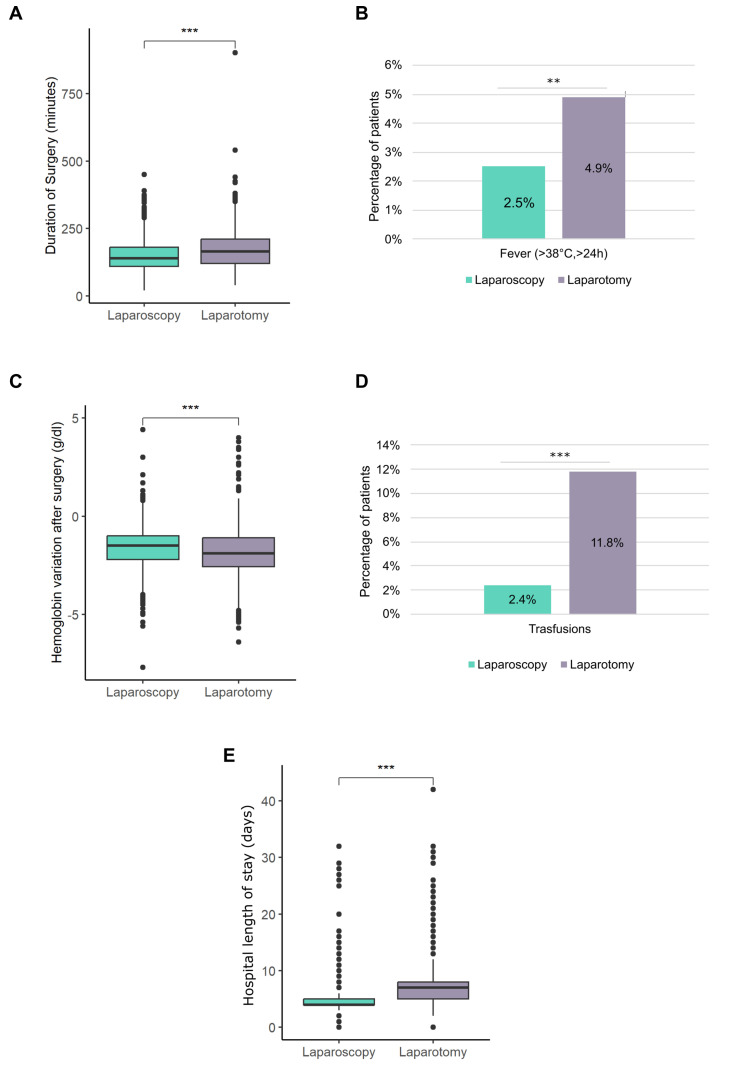
(**A**) Boxplots representing the duration of surgery in patients treated with laparoscopy or laparotomy. (**B**) Histograms illustrating the percentage of patients who had a fever higher than 38° for more than 24h after laparoscopy or laparotomy. (**C**) Boxplots representing the hemoglobin variation after surgery in patients treated with laparoscopy or laparotomy. (**D**) Histograms illustrating the percentage of patients who needed a transfusion after laparoscopy or laparotomy. (**E**) Boxplots summarizing the hospital length of stay required after a laparoscopy or laparotomy. ** *p* < 0.01, *** *p* ≤ 0.001.

**Figure 4 cancers-17-02261-f004:**
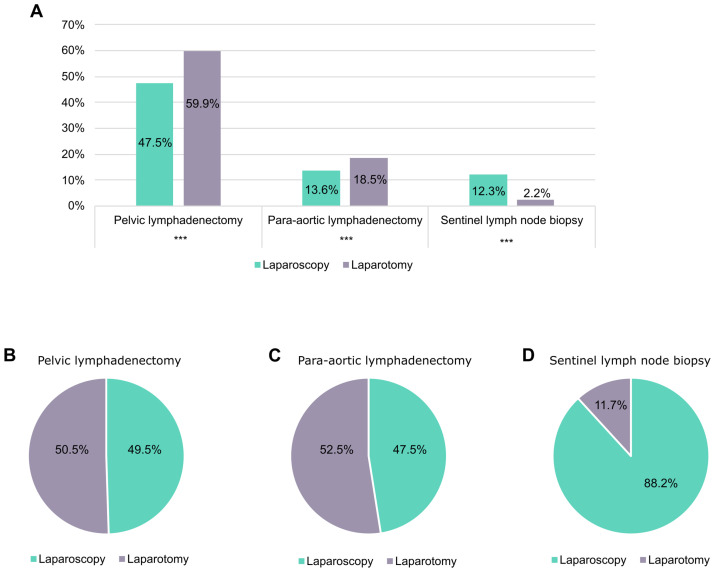
(**A**) Histograms representing the percentage of patients treated with laparoscopy or laparotomy who received pelvic, paraaortic lymph node dissection, or sentinel lymph node biopsy. (**B**) Pie chart of the percentages of laparoscopies and laparotomies performed in patients who needed pelvic lymph node dissection. (**C**) Pie chart of the percentages of laparoscopies and laparotomies performed in patients who needed paraaortic lymph node dissection. (**D**) Pie chart of the percentages of laparoscopies and laparotomies performed in patients who needed sentinel lymph node biopsy. *** *p* ≤ 0.001.

**Figure 5 cancers-17-02261-f005:**
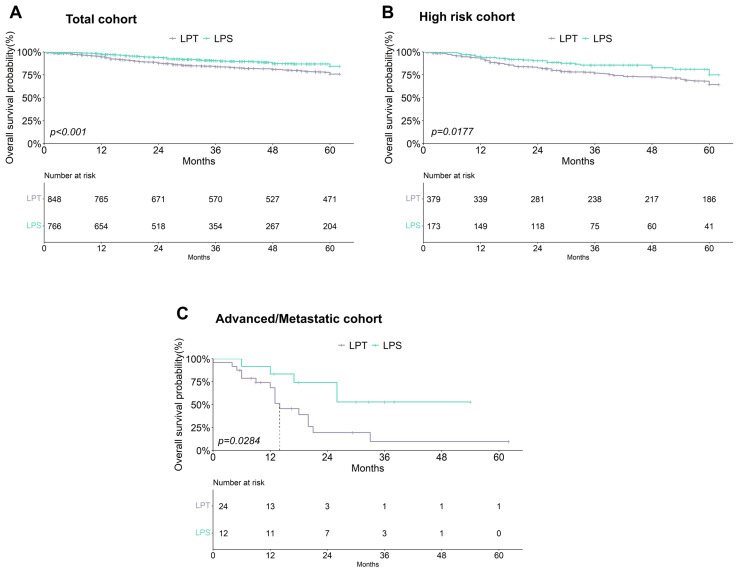
Kaplan–Meier curves comparing the effect of surgical procedures on overall survival in the total cohort of EC patients (**A**) and in high-risk (**B**) and metastatic/advanced (**C**) groups.

**Figure 6 cancers-17-02261-f006:**
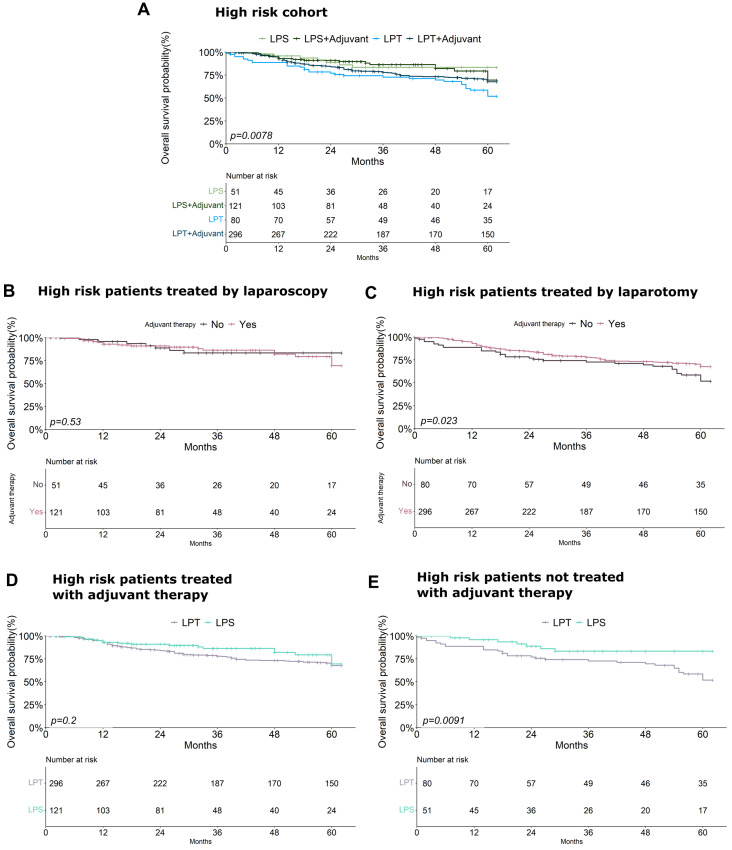
(**A**–**E**) Kaplan–Meier curves showing: (**A**) The combinatory effect of surgical treatment and adjuvant therapy on the overall survival of high-risk EC patients. (**B**) The effect of adjuvant therapy on the overall survival of high-risk EC patients treated with laparoscopy. (**C**) The effect of adjuvant therapy on the overall survival of high-risk EC patients treated with laparotomy. (**D**) The effect of surgical treatment on the overall survival of high-risk EC patients who received adjuvant therapy. (**E**) The effect of surgical treatment on the overall survival of high-risk EC patients who did not receive adjuvant therapy.

**Figure 7 cancers-17-02261-f007:**
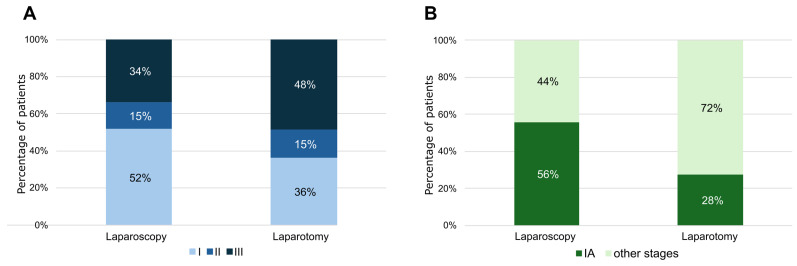
(**A**) Histograms showing the distribution of FIGO stages in EC patients treated with laparoscopy or laparotomy. (**B**) Histograms showing the distribution of IA stages in non-endometrioid EC patients treated with laparoscopy or laparotomy.

**Figure 8 cancers-17-02261-f008:**
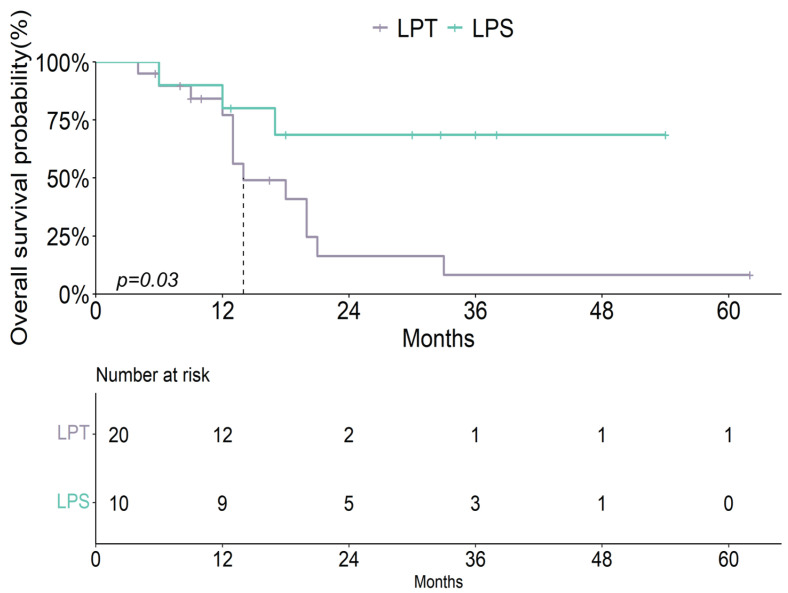
Kaplan–Meier curves showing the effect of surgical treatment in advanced/metastatic EC patients treated with adjuvant therapy.

**Table 1 cancers-17-02261-t001:** Summary of clinical data from 2402 EC patients.

	Overall (N = 2402)
**Age (years)**	
Median	66.0
IQR	15.0
N-Miss	9
**BMI category**	
Normal weight	531 (25.4%)
Overweight	672 (32.1%)
Obese	889 (42.5%)
N-Miss	310
**Hemoglobin variation (24 h) (g/dl)**	
Median	−1.6
IQR	1.4
N-Miss	298
**Fever (>38 °C, >24 h)**	
No	2117 (96.4%)
Yes	78 (3.6%)
N-Miss	207
**ASA score**	
I	159 (7.1%)
II	1225 (54.7%)
III	833 (37.2%)
IV	24 (1.1%)
N-Miss	161
**Parity**	
Median	1.0
IQR	1.0
N-Miss	159
**Menopausal state**	
No	114 (4.9%)
Yes	2212 (95.1%)
N-Miss	76
**Hypertension**	
No	1071 (47.3%)
Yes	1195 (52.7%)
N-Miss	136
**Diabetes**	
No	1877 (83.2%)
Yes	378 (16.8%)
N-Miss	147
**HRT**	
No	1901 (92.2%)
Yes	160 (7.8%)
N-Miss	341
**Tamoxifen**	
No	2124 (95.9%)
Yes	91 (4.1%)
N-Miss	187
**Other therapies**	
No	997 (52.4%)
Yes	906 (47.6%)
N-Miss	499
**Symptoms**	
No	121 (5.3%)
Yes	2149 (94.7%)
N-Miss	132
**Pre-surgical examinations**	
D&C	262 (11.8%)
Hysteroscopy	1844 (82.9%)
Both	113 (5.1%)
None	5 (0.2%)
N-Miss	178
**Diagnostic imaging**	
CT	1626 (73.6%)
USG	25 (1.1%)
MR	333 (15.1%)
MR+CT	195 (8.8%)
XR	26 (1.2%)
PET-CT	4 (0.2%)
N-Miss	193
**Transfusion**	
No	2018 (93.7%)
Yes	136 (6.3%)
N-Miss	248
**Surgical approach**	
Vaginal hysterectomy	72 (3.0%)
LPS	1283 (53.6%)
LPT	1037 (43.4%)
N-Miss	10
**Peritoneal Biopsies**	
No	1858 (78.3%)
Yes	516 (21.7%)
N-Miss	28
**Peritoneal washing**	
No	783 (32.9%)
Yes	1595 (67.1%)
N-Miss	24
**PLND**	
No	1165 (48.6%)
Yes	1231 (51.4%)
N-Miss	6
**PALND**	
No	2029 (84.7%)
Yes	367 (15.3%)
N-Miss	6
**SLD biopsy**	
No	1225 (92.3%)
Yes	102 (7.7%)
N-Miss	1075
**Total number of excised lymph nodes**	
Median	13.0
IQR	22.0
**Percentage of positive lymph nodes**	
Mean (SD)	3.5 (12.5)
**Adnexectomy**	
Monolateral	18 (0.8%)
No	48 (2.0%)
Yes	2329 (97.2%)
N-Miss	7
**Duration of surgery (minutes)**	
Median	150.0
IQR	80.0
N-Miss	197
**LoS (days)**	
Median	5.0
IQR	3.0
N-Miss	181
**FIGO stage**	
I	1910 (81.0%)
II	106 (4.5%)
III	298 (12.6%)
IV	44 (1.9%)
N-Miss	44
**Histology**	
Endometrioid	2029 (84.7%)
Other histotypes	366 (15.3%)
N-Miss	7
**Grade (only endometrioid EC)**	
G1-G2	1696 (83.8%)
G3	328 (16.2%)
N-Miss	378
**LVSI**	
No	1617 (76.5%)
Yes	496 (23.5%)
N-Miss	289
**ESMO-ESGO Class of Risk**	
Low	912 (40.9%)
Intermediate	279 (12.5%)
High-Intermediate	290 (13.0%)
High	707 (31.7%)
Advanced/Metastatic	44 (2%)
N-Miss	170
**Adjuvant treatment**	
No	1341 (55.8%)
Yes	1061 (44.2%)
N-Miss	0
**Adjuvant treatment**	
None	1341 (56.1%)
BRT	327 (13.7%)
EBRT	138 (5.8%)
BRT+RT	205 (8.6%)
CHT	97 (4.1%)
CHT+BRT/EBRT	284 (11.9%)
N-Miss	10
**Recurrence**	
No	2157 (89.8%)
Yes	245 (10.2%)
N-Miss	0
**Site of recurrence**	
Abdominal	76 (32.3%)
Extra abdominal	95 (40.4%)
locoregional	64 (27.2%)
N-Miss	10
**Death**	
No	1843 (80.3%)
Yes	451 (19.7%)
N-Miss	108
**Total survival (months)**	
Median	46.5
IQR	60.0
N-Miss	720
**Disease free survival (months)**	
Median	48.0
IQR	53.6
N-Miss	737

N-Miss: missing data, IQR: interquartile range, BMI: body mass index, ASA: American Society of Anesthesiologists, HRT: hormone replacement therapy, D&C: dilation and curettage, CT: computed tomography, USG: ultrasonography, MR: magnetic resonance, XR: X-ray, PET: positron-emission tomography, LPS: laparoscopy, LPT: laparotomy, PLND: pelvic lymph node dissection, PALND: paraaortic lymph node dissection, SLD: sentinel lymph node, LoS: hospital length of stay, FIGO: International Federation of Gynaecology and Obstetrics, LVSI: lymphovascular space invasion, ESMO: European Society for Medical Oncology, ESGO: European Society of Gynaecological Oncology, BRT: brachytherapy, EBRT: radiotherapy, CHT: chemotherapy. Other therapies are statins, antiarrhythmics, inotropes, antiaggregants, anticoagulants, antiosteoporotics, psycholeptics, antiparkinsonians, antidepressants, control of hyperuricemia, ursodeoxycholic acid, immunosuppressants, and antivirals.

**Table 2 cancers-17-02261-t002:** Univariate Cox analysis of the impact of surgical strategy on overall survival. LPT = laparotomy, LPS = Laparoscopy, CI.95 = 95% confidence interval.

	Patients(n)	Events(n)	Surgery	Hazard Ratio	CI.95	*p* Value
Total cohort	848	177	LPT	Ref		-
	766	73	LPS	0.58	[0.44;0.76]	<0.001
Low-risk cohort	229	16	LPT	Ref		
	312	22	LPS	1.57	[0.82;3.02]	0.172
Intermediate-risk cohort	112	15	LPT	Ref		
	65	8	LPS	1.18	[0.50;2.79]	0.708
Intermediate–high-risk cohort	67	8	LPT	Ref		
	103	9	LPS	0.86	[0.33;2.23]	0.754
High-risk cohort	379	117	LPT	Ref		
	173	26	LPS	0.60	[0.39;0.91]	0.0177
Advanced/metastatic cohort	24	16	LPT	Ref		
	12	5	LPS	0.32	[0.11;0.89]	0.0284

## Data Availability

Data is available from the authors upon reasonable request.

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
