# Peer review of "Impact of a Surgical Approach on Endometrial Cancer Survival According to ESMO/ESGO Risk Classification: A Retrospective Multicenter Study in the Northern Italian Region"

_cancers, 2025, doi:10.3390/cancers17132261_

Round 1
Reviewer 1 Report
Comments and Suggestions for Authors
From a biostats and clinical epidemiology point of view, here are some comments for the authors
- abstract: LACC, EC, FIGO, ESMO-ESGO and so on, define all of them properly
- line 50 319605 new cancer cases and 76160 cancer deaths, give updated estimates, i.e. those coming from the GBD2021 study (https://www.thelancet.com/gbd)
- line 77 which shocked the scientific community, give some quick details about this shock!
- line 88 seven hospitals, which ones?
- table 1 and everywhere along the text, describe continuous covariates only as median/IQR
- table 1 add measure units (i.e. Hb, LoS and so on)
- table 1 Other therapies, which ones?
- line 124 use a non-parametric inferential approach, ok for Fisher, while Kruskal-Wallis for Anova and so on
- line 126 I warmly recommend to estimate a Cox PH regression model too, both uni- and multi- variable; the lackness of these models is a weak point of this manuscript
- line 126 add the median follow-up, both for the whole cohort and the LPS/LPT subgroups
Author Response
Dear reviewer,
Thank you for your time and suggestions that we followed to improve our article. We hope that it can now be accepted for publication.
Best regards
Reviewer 1
From a biostats and clinical epidemiology point of view, here are some comments for the authors
- abstract: LACC, EC, FIGO, ESMO-ESGO and so on, define all of them properly
Reply
All abbreviations have been reported in extenso
- line 50 319605 new cancer cases and 76160 cancer deaths, give updated estimates, i.e. those coming from the GBD2021 study (https://www.thelancet.com/gbd)
Reply
We have updated epidemiological data as requested: . In 2022, 420,242 new cases of cancer were diagnosed and 97,704 cancer deaths occurred worldwide [Bray F, Laversanne M, Sung H, Ferlay J, Siegel RL, Soerjomataram I, Jemal A. Global cancer statistics 2022: GLOBOCAN estimates of worldwide incidence and mortality for 36 cancers in 185 countries. CA Cancer J Clinic. 2024 May-Jun;74(3):229-263. doi: 10.3322/caac.21834.]
- line 77 which shocked the scientific community, give some quick details about this shock!
Reply
We briefly reported the results of the LACC trial and explained the doubts that arose from them.
“Cervical cancer patients treated with minimally invasive surgery presented a higher recurrence rate and worse overall survival than patients treated with LPT [20], did EC patients treated with LPS also have the same risk?”
- line 88 seven hospitals, which ones?
Reply
We have listed the hospitals that took part in the study:
Azienda USL-IRCCS di Reggio Emilia, Reggio Emilia, Italy. University of Bologna, Bologna, Italy. University of Modena and Reggio Emilia, Modena, Italy. University of Parma, Parma, Italy. University of Ferrara, Ferrara, Italy. Ospedale di Forlì, Forlì, Italy. Ospedale degli Infermi, Rimini, Italy.
- table 1 and everywhere along the text, describe continuous covariates only as median/IQR
Reply
As requested by the reviewer, we brought continuous covariates as median/IQR. See new table 1 and the revised manuscript.
- table 1 add measure units (i.e. Hb, LoS and so on)
Reply
Measure units were reported in new table 1.
- table 1 Other therapies, which ones?
Reply
Other therapies have been reported in the legend of table 1: statins, antiarrhythmics, inotropes, antiaggregants, anticoagulants, antiosteoporotics, psycholeptics, antiparkinsonians, antidepressants, control of hyperuricemia, ursodeoxycholic acid, immunosuppressants, antivirals
- line 124 use a non-parametric inferential approach, ok for Fisher, while Kruskal-Wallis for Anova and so on
Reply
Statistical analysis were performed as requested
- line 126 I warmly recommend to estimate a Cox PH regression model too, both uni- and multi- variable; the lackness of these models is a weak point of this manuscript
Reply
The Cox regression model was run as required
- line 126 add the median follow-up, both for the whole cohort and the LPS/LPT subgroups
Reply
Median follow-up data were reported :
“The analysis was conducted using all available data without imputation. For each variable, the number of missing observations was documented. (Table 1). Total cohort presented a median follow up period of 46.5 months (IQR=60 months), sub-cohort treated with laparoscopy presented a median follow up of 33 months (IQR=40 months) while sub-cohort treated with laparotomy presented a median follow up of 64 months (IQR=88.5 months).”

Reviewer 2 Report
Comments and Suggestions for Authors
- Table 1 contains a lot of missing data ( N - Miss ), which could affect the final results and their reliability. Recommended In the Materials and Methods section, add information on how missing data were handled.
- All the figures in the manuscript are overloaded with data, which makes it difficult to perceive what is important information. It is recommended to split the drawings into several separate images with a detailed interpretation of the results.
- In Figure 3 and 4 very small font, which reduces readability. It is recommended to increase the font size and improve the quality of the images.
- Uneven distribution of LPS patients vs. LPT . Patients who underwent LPS were younger and were less likely to have concomitant diseases (hypertension, diabetes, etc.) than patients who underwent LPT . This suggests a systematic selection of less severe patients for LPS , which may have biased the results in its favour. It is recommended to state in the limitations that group imbalance may have influenced the results.
- In the survival analysis in Figures 3-4 key prognostic factors (stage, histology, LVSI , adjuvant therapy) were not taken into account. Better survival in the high-risk LPS group (Figure 3 E ) may not be explained by the type of procedure performed operations, and a lower prevalence of the disease (fewer patients with stage III ).
- Survival improved in the laparoscopic group in high - risk patients without adjuvant therapy – this finding requires careful interpretation. It is possible that those who did not receive adjuvant therapy and were operated laparoscopically had a lower risk according to histological criteria. Recommendation : detail characteristic these patients .
- The statement “In the literature, the use of LPS varies from 33.6% to over 80% of EC high-volume hospitals” is repeated twice . lines 284-287.
- For a more objective understanding of the problem, it would be advisable to include in the discussion the article : Ramirez P.T., Robledo K.P., Frumowitz M., Pareja R., Ribeiro R., Lopez A., Yang H., Isla D., Moretti R., Bernardini M.K., Gebsky V., Asher R., Behan V., Coleman R.L., The Obermair A. LACC study: A definitive analysis of overall survival when comparing Open and minimally invasive radical hysterectomy for early-stage cervical cancer. J Clinical oncology. 2024, August 10;42(23):2741-2746. doi : 10.1200/JCO.23.02335. Published on May 29, 2024. PMID: 38810208.
Author Response
Dear reviewer,
Thank you for your time and suggestions that we followed to improve our article. We hope that it can now be accepted for publication.
Best regards
Reviewer 2
- Table 1 contains a lot of missing data ( N - Miss ), which could affect the final results and their reliability. Recommended In the Materials and Methods section, add information on how missing data were handled.
Reply
Information about missing data were reported in the Materials and Methods section:
“The analysis was conducted using all available data without imputation. For each variable, the number of missing observations was documented”.
- All the figures in the manuscript are overloaded with data, which makes it difficult to perceive what is important information. It is recommended to split the drawings into several separate images with a detailed interpretation of the results.
Reply
All figures have been revised and improved.
- In Figure 3 and 4 very small font, which reduces readability. It is recommended to increase the font size and improve the quality of the images.
Reply
Figures have been revised and improved.
- Uneven distribution of LPS patients vs. LPT . Patients who underwent LPS were younger and were less likely to have concomitant diseases (hypertension, diabetes, etc.) than patients who underwent LPT . This suggests a systematic selection of less severe patients for LPS, which may have biased the results in its favour. It is recommended to state in the limitations that group imbalance may have influenced the results.
Reply
To better understand the effect of therapeutic choices on overall survival (OS) and in accordance with previous analyses, multivariate Cox regression models were applied to the total cohort. Two models were constructed: Model 1, which included risk class, surgical strategy, and adjuvant therapy; and Model 2, which added the variables age, diabetes, hypertension and ASA score to evaluate its their potential confounding effect on the associations between clinical variables and survival outcomes. As expected, age emerged as a significant negative prognostic factor for endometrial cancer. Regarding surgical approach, LPS was significantly associated with improved survival in Model 1 (HR 0.74, p = 0.0435), but this association was weakened and became non-significant in Model 2 (HR 0.7876, p = 0.09860750).
Similar dual models were constructed to assess OS in two sub-cohorts: high-risk patients and ad-vanced/metastatic patients. For the high-risk group, the models also included FIGO stage, histological subtype, and LVSI. For the advanced/metastatic group (entirely FIGO stage IV), histotype and LVSI were added. In the high-risk sub-cohort, age remained a significant negative prognostic factor. The surgical approach showed a consistent and robust protective effect of LPS compared to LPT in both models, with an approximately 50% reduction in hazard (Model 1 HR = 0.50, p = 0.0058; Model 2 HR = 0.5248, p = 0.0042104), indicating that the survival benefit of LPS in this subgroup is independent of age.
- In the survival analysis in Figures 3-4 key prognostic factors (stage, histology, LVSI , adjuvant therapy) were not taken into account. Better survival in the high-risk LPS group (Figure 3 E ) may not be explained by the type of procedure performed operations, and a lower prevalence of the disease (fewer patients with stage III ). Survival improved in the laparoscopic group in high - risk patients without adjuvant therapy – this finding requires careful interpretation. It is possible that those who did not receive adjuvant therapy and were operated laparoscopically had a lower risk according to histological criteria. Recommendation : detail characteristic these patients .
Reply
As reported in the results section, the distribution of FIGO stages was significantly different (p < 0.001) between high-risk patients treated with LPS or LPT: among patients treated with LPS 51% were at stage I, 15% at stage II and 34% at stage III, while in the LPT group 37% were at stage I, 15% at stage II and 48% at stage III (Fig. 7A). Furthermore, focusing on non-endometrioid ECs, in the LPS-treated group 56% of ECs were stage IA versus 28% in the LPT-treated group (p < 0.001) (Figure 7B).
In multivariate Cox regression models were applied to high-risk patients. The models also included FIGO stage, histological subtype, and LVSI. As expected, FIGO stage III was significantly associated with worse OS compared to stage I, whereas stage II was not significantly different. Similarly, non-endometrioid histology was significantly associated with worse survival. However, the surgical approach showed a consistent and robust protective effect of LPS compared to LPT, with an approximately 50% reduction in hazard, indicating that the survival benefit of LPS in this subgroup is independent. On contrary, adjuvant therapy was not significantly associated with OS in the high-risk subgroup when adjusted in multivariate models. In the advanced/metastatic sub-cohort the only variable consistently associated with a strong and significant improvement in OS was adjuvant therapy indicating a robust protective effect of this treatment in patients with advanced disease (Table S2). However, in the discussion session, we reported that in cases where serous tumors are limited to polyps or are intramucosal tumors, if patients are well staged, adjuvant therapy could be avoided.
- The statement “In the literature, the use of LPS varies from 33.6% to over 80% of EC high-volume hospitals” is repeated twice . lines 284-287.
Reply
The repeated phrase has been deleted
- For a more objective understanding of the problem, it would be advisable to include in the discussion the article : Ramirez P.T., Robledo K.P., Frumowitz M., Pareja R., Ribeiro R., Lopez A., Yang H., Isla D., Moretti R., Bernardini M.K., Gebsky V., Asher R., Behan V., Coleman R.L., The Obermair A. LACC study: A definitive analysis of overall survival when comparing Open and minimally invasive radical hysterectomy for early-stage cervical cancer. J Clinical oncology. 2024, August 10;42(23):2741-2746. doi : 10.1200/JCO.23.02335. Published on May 29, 2024. PMID: 38810208.
Reply
The article was added to the discussion:
“The final analysis of the results of the LACC trial confirmed that worse survival was associated with the minimally invasive approach; furthermore, patients treated with the minimally invasive approach more often presented peritoneal carcinomatosis at recurrence [52]”.

Reviewer 3 Report
Comments and Suggestions for Authors
This in an interesting study. However, it should be clearly mentioned that LAP2 data showed similar survival for patients after laparoscopy or laparotomy and the same was true for five-year recurrence rates.
Author Response
Dear reviewer,
Thank you for your time and suggestion that we followed to improve our article. We hope that it can now be accepted for publication.
Best regards
Reviewer 3
This in an interesting study. However, it should be clearly mentioned that LAP2 data showed similar survival for patients after laparoscopy or laparotomy and the same was true for five-year recurrence rates.
Reply
We underlined that the result was similar to that of the LAP2 study

Round 2
Reviewer 1 Report
Comments and Suggestions for Authors
The Authors were able to solve all previous concerns